# Phase-Based Grasp Classification for Prosthetic Hand Control Using sEMG

**DOI:** 10.3390/bios12020057

**Published:** 2022-01-21

**Authors:** Shuo Wang, Jingjing Zheng, Bin Zheng, Xianta Jiang

**Affiliations:** 1Department of Computer Science, Memorial University of Newfoundland, St. John’s, NL A1C 5S7, Canada; shuow@mun.ca (S.W.); jzheng20@mun.ca (J.Z.); 2Department of Surgery, University of Alberta, Edmonton, AB T6G 2R3, Canada; bzheng1@ualberta.ca

**Keywords:** myoelectric prosthesis, sEMG, grasp phases analysis, grasp classification, machine learning

## Abstract

Pattern recognition using surface Electromyography (sEMG) applied on prosthesis control has attracted much attention in these years. In most of the existing methods, the sEMG signal during the firmly grasped period is used for grasp classification because good performance can be achieved due to its relatively stable signal. However, using the only the firmly grasped period may cause a delay to control the prosthetic hand gestures. Regarding this issue, we explored how grasp classification accuracy changes during the reaching and grasping process, and identified the period that can leverage the grasp classification accuracy and the earlier grasp detection. We found that the grasp classification accuracy increased along the hand gradually grasping the object till firmly grasped, and there is a *sweet period* before firmly grasped period, which could be suitable for early grasp classification with reduced delay. On top of this, we also explored corresponding training strategies for better grasp classification in real-time applications.

## 1. Introduction

Losing a hand is a tremendously physical trauma to any individual. Amputated individuals face a huge difficulty in performing daily activities independently [1], which can also lead to unemployment and social isolation [2]. According to statistics, only 66% of them can resume work after that [3].

To restore the functionality of hands in daily life and in work place, wearing prostheses is one of the necessary options for amputees. There are three types of prosthetic hands: cosmetic hand, body-power hand and Myoelectric hand [4,5]. Among them, Myoelectric prosthesis hand is the most promising one, which allows an amputee to controls the robotic hand by reading his/her muscle actives using Surface Electromyography (sEMG) sensors on the residual forearm. The computer chip will read in muscle signals and convert signals into executable commands.

Interpretation on muscle signals is essential for the control of electric powered prosthetic hands, which requires machine learning algorithms to classify muscular electric signals into corresponding hand movement patterns. In most of the published papers, scientists use myoelectric signals recorded during firmly grasped periods for grasp classification, which yielded satisfactory classification outcomes [6,7,8,9,10,11]. For instance, the research done by Jiang et al. [7] using 3 s firm grasp sEMG signals achieved approximately 85% accuracy for classifying 16 grasp gestures. However, the firmly grasped periods occur at the end of reaching and grasping, giving no time to control arm movement in a real-life environment [12].

When including muscle activities recorded from entire grasp period, the classification accuracy decreased. In Cognolato et al.’s report [13], the accuracy of the classification for 10 grasp gestures was approximately 63% to 82% by using the sEMG signals during the whole grasp period.

To solve this problem, developing a method to classify grasp pattern using sEMG data recorded in the earlier grasp period with a high accuracy is necessary. In this study, we investigate how grasp classification accuracy changes over the entire reaching and grasping process, and identify a period in early grasp phase that can achieve the best grasp classification outcome. We call this period as *sweet period*. Once the sweet period is identified, we can develop a better classification strategy that can be used in the real-time environment.

Specifically, we first apply and compare several processing methods for the feature extraction of the sEMG signals. Then, we conduct an experiment to find the sweet period that is suitable for early grasp classification with the best classification outcome. Finally, we will conduct another experiment to compare several common training and testing strategies to identify an effective strategy for better real-time grasp classification. We hypothesize that the muscle activities recorded in the early period of hand grasping can provide sufficient information to achieve the same or higher accuracy of grasp classification than other time periods with a reduced delay for prosthetic hand control.

## 2. Materials and Methods

### 2.1. Data Collection

The data used in this study were from an open-source dataset collected by Cognolato et al. [13], where the sEMG data were recorded from 30 healthy subjects (27 male and 3 female), with an average age of 46.63 ± 15.11 years.

Twelve sEMG sensors were placed on the forearm of each subject, producing twelve columns of sEMG data, respectively. Due to the hardware problem, no myoelectric data were received from electrode number eight during the acquisition of subject S024. Therefore, the sEMG data for this subject were recorded from eleven electrodes instead of twelve [13].

Ten grasp gestures were performed in this data collection which were selected based on the hand taxonomies [14,15,16,17] and grasp frequency in Activities of Daily Living [18]. The participant performed each gesture for four repetitions, and in each repetition, the same gesture was performed three times using three different objects, respectively. A designated experimenter vocally guided the participant to perform which gestures and grasp which objects. The data were labelled according to the vocal instruction. The list of gestures and objects are shown in Table 1.

In the data post-processing part, the abnormal samples were replaced with the precedent valid samples when filtering outliers [13]. As there might be a delay between the participants’ response to the vocal instructions [13], the sEMG activation time might not be matched perfectly with the stimulus time. Therefore, relabeling was performed to calibrate this difference using the method described by Kuzborskij et al. [19].

### 2.2. Electromyography Feature Extraction and Selection

In the feature extraction process, we first determined the suitable window size for deriving features [20]. As shown in Table 2, several sizes of the overlapped window were tested, which are 50 ms, 100 ms, 200 ms, 500 ms, and 1000 ms. As the increase of the window size, the accuracy keeps increasing, which means that the more data we used to derive features, the better performance we could get. However, considering the capability of Myoelectric prosthesis in the real-life condition, a large window would delay the grasp action from the prosthetic hand. On the other hand, it can be seen that, when increasing the window size over 200 ms, the increase of the accuracy is less than 1%, which is a very small increase. Therefore, to keep the balance between accuracy and implementation speed, we chose 200 ms as the window length with the step of 50 ms, which is a 75% overlap between successive windows.

To assure the recognition accuracy by using proper features, we tested eleven commonly used features, which were Standard Deviation (STD), Root Mean Square (RMS), Integrated EMG (IEMG), Mean Absolute Value (MAV), Waveform Length (WL), Log Detector (LOG), Simple Square Integral (SSI), Skewness (SKW), Kurtosis (KURT), Average Amplitude Change (AAC) and Difference Absolute Standard Deviation Value (DASDV) [21]. We dropped three lowest performance features, whichwere LOG, SKW and KURT and chose the rest eight with the highest accuracy as the final features for the following research. The performance of these features are shown in Figure 1. After applying the eight features to the sEMG signals, the data set was converted from 12 columns to 96 columns. Due to the sensor hardware issue mentioned in the first subsection, the sEMG data of subject S024 was changed from 11 columns to 88 columns.

### 2.3. Classification Models

Gradient boosting decision tree, such as XGBoost [22] and Light Gradient Boosting Machine (LightGBM) [23], is a popular machine learning algorithm used by a large amount of data scientists recently, which can achieve a high performance by using decision trees as weak learners and assembling them to come up with one strong learner. Considering the high feature dimensions and large data size, we chose LightGBM as the classifier which runs faster while maintaining a high level of accuracy by utilizing two novel techniques called Gradient-Based One-Side Sampling (GOSS) and Exclusive Feature Bunding (EFB) [23]. In the experiment of Ke et al. (2017), LightGBM can accelerate the training process up to twenty times than XGBoost.

We tuned the hyperparameters by using the training set of all the subjects and obtained the best results as follows: the learning rate is 0.1; no limit was set for the maximum depth; the number of estimators is 100; the number of leaves is 31; the remaining parameters are set to the default values.

### 2.4. Phase-Based Grasp Analysis

Normally, a typical reaching and grasping process can be divided into three phases [24,25]:The Reaching Phase: starts from the hand lifting off, and ends by touching the object. During this phase, the hand is accelerated to a peak velocity and then is decelerated and brought to touch the target object. The hand usually opens to be configured to the target grasp gesture (pre-shape) [26].The Early Grasping Phase: begins at the moment when the hand initially contacts the object, and gradually closes the fingers until the hand starts to firmly grasp the object.The Firm Grasping Phase: the target object is firmly grasped and hand shape is maintained relatively steady.

We segmented the Reaching, Early grasping, and Firm Grasping phases of each grasp gesture from each subject by observing corresponding videos frame by frame and calculated the average duration of each phase from all the observations. The judgment criteria for entering an Early Grasping Phase was the moment that the hand started to touch the target object, the judgement criteria for entering a Firm Grasping Phase was the moment that the target grasp gesture was completely formed and the hand started to keep relatively steady. According to the segmentation, Early Grasping Phase and Firm Grasping Phase started averagely 1020 ms and 1604 ms from the beginning of Reaching Phase, respectively. An example of grasp phases overlaid with sEMG signals during a full grasp trail is shown in Figure 2.

## 3. Experiments and Results

We conducted two experiments, the first one aimed to analyse the grasp classification accuracy during the three grasping phases and find out the best position and length of sweet period, another was to find out the best training strategy.

### 3.1. Data Processing

The grasp trials performed by the participants lasted approximately 4.5–5 s [13]. We removed the data after 4.5 s to align all the trials the same length. Because the overlapped window step is 50 ms and the grasp period length is 4.5 s, 90 pieces of data were reminded for each trial.

In this study, each participant performed one grasp gesture four times (repetitions) which allowed us to split the sEMG data by repetitions to validate testing results. For all the cases in this study, we used three repetitions (75%) for training and one repetition (25%) for testing with leave-one-repetition-out cross-validation, which used one repetition data for testing and the rest three repetitions for training the model, and repeated this process four times to cover all repetitions for testing. To increase the reliability of the sEMG data set, there were three objects being grasped in each repetition with the same gesture as mentioned in the data collection section. In other words, there were 324,000 data samples (90 samples/grasp × 10 grasp gestures × 4 repetitions × 3 objects × 30 subjects) in the data set.

### 3.2. Phases and Sweet Period Analysis

Figure 3 shows the mean changes of testing accuracy of grasp classification during all the three phases. Each data point is averaged across all 900 trials from 30 participants.

From Figure 3 we can see that the accuracy increases from 42% to 84% during the Reaching phase and then becomes stable at the start of Early Grasping phase at around the time of 1000 ms, fluctuating between 84% and 87% during the rest of the grasp period. The mean accuracy further increases to relatively stable at around the time of 1250 ms, where we then define the location of the sweet period.

To find the optimal length of the sweet period, we designed different sliding windows with sizes of 300 ms, 400 ms, 500 ms, 600 ms, 700 ms, 800 ms, 900 ms and 1000 ms. The sliding window moved along with the time with step 50 ms, and in each move, it calculated and recorded the mean accuracy. We analyzed the records from the sliding window, and the results are given in Figure 4.

From Figure 4, we can see that the mean accuracy increases with the increase of window length significantly during the Reaching phase and beginning of the Early Grasping phase (at about 1100 ms) but not significantly afterward. For instance, although the window length of 1000 ms can reach the highest accuracy of 86.3%, it takes a much longer time than the length of 300 ms with an accuracy of 85.5%. Therefore, the length of the sweet period is set as 300 ms, and the position is set from 1100 ms to 1400 ms, which makes it entirely located in the Early Grasping phase as the blue region shown in Figure 3.

### 3.3. Comparison Experiment

In the comparison experiment, we tested six strategies using different training and testing data, as shown in Table 3.

In cases 1–3, we used all the three grasp phases as training data and reduced the testing data size, from all three phases to only the firm grasping phase, then to the sweet period. The purpose of performing these three comparisons was to study which phase/period was the better choice for testing data when using all grasp phases as training data. Besides, to figure out which phase played a better role model training, we studied another five cases. For cases 4–5, we used Firm Grasping Phase for training and reduced the testing data size. In cases 6–7, we used a combined Phases for training and sweet period for testing. In case 8, we used the data in the sweet period for both training and testing. It is worth mentioning again, the cross-validation method used for all the cases was leave-one-repetition-out cross-validation which used one repetition data for testing and the rest three repetitions for training the model, and repeated this process four times to cover all repetitions for testing, such that all testing data was excluded from training the model. For example, in one testing repetition of case 8, the data from the sweet period of three repetitions were used for training the model and the rest one for testing. The results are presented in Table 3.

As shown in the Table 3, we get the highest accuracy of 85.50% when we train with the all grasp phases and test with the only sweet period. Besides, from case 1 to 3, we find that if we keep the training data unchanged, the accuracy increases as the decrease of testing data size.

## 4. Discussion

Our hypothesis is supported by the results that there is a sweet period located in the Early Grasping Phase where sEMG signals can be used to achieve a similar or higher accuracy and lower delay of grasp classification than other time windows, which would help to improve the performance of robotic hand implementation in the real-life applications. This is important as the classifier can get the data much faster instead of waiting the muscle getting into the Firm Grasping Phase.

We found that during the Reaching Phase, the mean accuracy of this phase is only about 63%. This is because, in this period, the subjects moved their hands to reach the object and start to perform the grasp gesture, keeping the muscle status changing. Therefore, the sEMG signals in this period fluctuate very much, making it difficult in decoding the sEMG signals, see Figure 2.

When getting into the Early Grasping phase, the accuracy reaches approximately 85%, which is as high as that in the Firm Grasping phase. The possible reason for this is the hand has already fully formed into the target gesture during the Early Grasping phase. Although this formed gesture is slightly different to the final target gesture, it can provide sufficient information for the classification. Therefore, the accuracy reaches to a high level at the start of the Early Grasping phase. After the subject firmly grasps the object (getting into the Firm Grasping phase), the accuracy keeps stable at around 85% because the sEMG signals started to be stable, which also make the classification performance stable.

Notice that the sEMG signal is more active in the Reaching and Early Grasping phases with high amplitude of the sEMG waveform as shown in Figure 2. This is because the hand starts to perform the corresponding grasping gestures related activities such as hand aperture, where the sEMG signals from the forearm are usually active with higher amplitude than other phases [26], although the hand has not grasped to the object during the Reaching Phase. In contrast, starting from the mid-Early Grasping phase to the whole Firm Grasping Phase, the muscle status keeps relatively unchanged, which makes the amplitude sEMG signal slightly lower than that in the reaching and grasping phase; this is also why better grasp classification performance was achieved during the Early Grasping phase and the Firm Grasping Phase where the sEMG signal patterns are relatively similar.

Using all three grasp phases for training the model and only using sweet period for controlling is found to be the best strategy for Myoelectric prosthetic hand application in real-life condition, not only because the sweet period during Early Grasping phase is suitable for prosthesis control as discussed before, but also this strategy can also increase the recognition accuracy compared to other strategies. The possible reason of higher accuracy achieved by this strategy could be more variation data was included in the model training. From case 3 and 6 in the Table 3, we can see that if we remove the Firm Grasping Phase from training set, the accuracy decreases from 85.5% to 81.01%. This means that the Firm Grasping Phase is essential for training data because it may contain the information about the final target gesture. From case 3 and 7, we find that if we remove Reaching Phase from training set, the accuracy decreases from 85.5% to 82.51%. This means that Reaching Phase is also important for training data because it is the progress in which the gesture is formed.

For case 5, the accuracy is only 60.80% when only using the Firm Grasping Phase for training because this period lost much information about gesture formation in Reaching and Early Grasping Phases. For case 8, the accuracy reaches 74.99% only using sweet period for training because this training data also lost the part of information about the gesture in the Reaching Phase and the Firm Grasping Phase. However, using all phases data for training and the sweet period data for testing achieved the best accuracy, which can be the common practice in real-life situations where training a model is not time-sensitive.

## 5. Conclusions

In order to reduce the delay of myoprosthetic hand control in the real-life situation while maintaining a high recognition accuracy, we investigated the grasp classification performance during three grasping phases to identify the sweet period. We found that the sweet period located between 1.1 s and 1.4 s from the start of the hand grasping which happens in the Early Grasping phase before the hand is firmly grasped.

Furthermore, we found using sEMG from all three grasping phases (Reaching, Early Grasping, and Firm Grasping phases) for grasp classification model training achieved the best accuracy. Together with the identified sweet period for controlling, the grasp classification accuracy and the response speed of prosthetic hand can be balanced to achieve high performance. 

## Figures and Tables

**Figure 1 biosensors-12-00057-f001:**
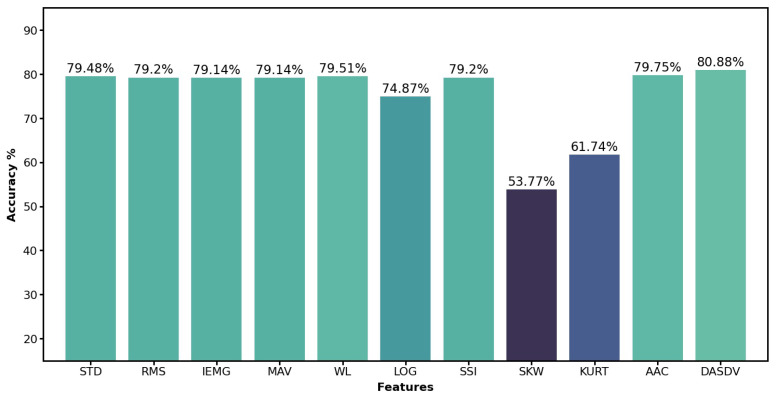
Single feature performance with window size 200 ms. The eleven features are Standard Deviation (STD), Root Mean Square (RMS), Integrated EMG (IEMG), Mean Absolute Value (MAV), Waveform Length (WL), Log Detector (LOG), Simple Square Integral (SSI), Skewness (SKW), Kurtosis (KURT), Average Amplitude Change (AAC) and Difference Absolute Standard Deviation Value (DASDV). The classifier used was lightGBM. The cross-validation method used was leave-one-repetition-out cross-validation which used one repetition data for testing and the rest three repetitions for training the model, and repeated this process four times to cover all repetitions for testing.

**Figure 2 biosensors-12-00057-f002:**
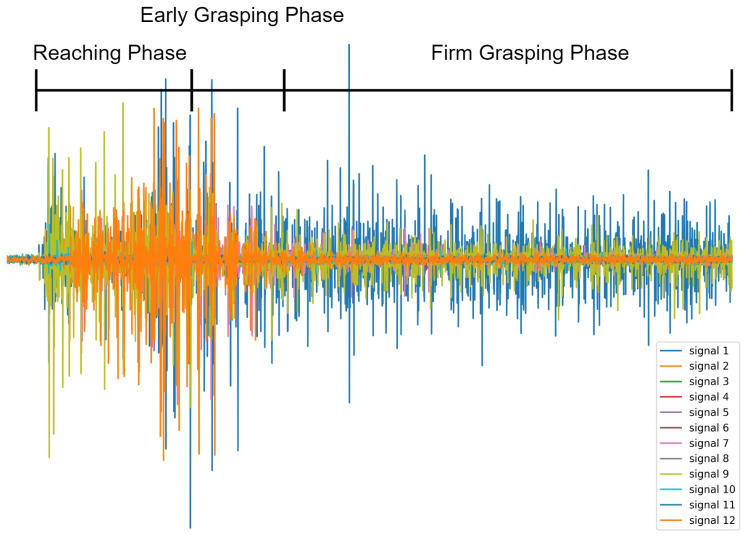
An example of grasp phases overlaid with sEMG signals during a full grasp trial. The start and end positions of these three phases were determined by observing corresponding videos frame by frame.

**Figure 3 biosensors-12-00057-f003:**
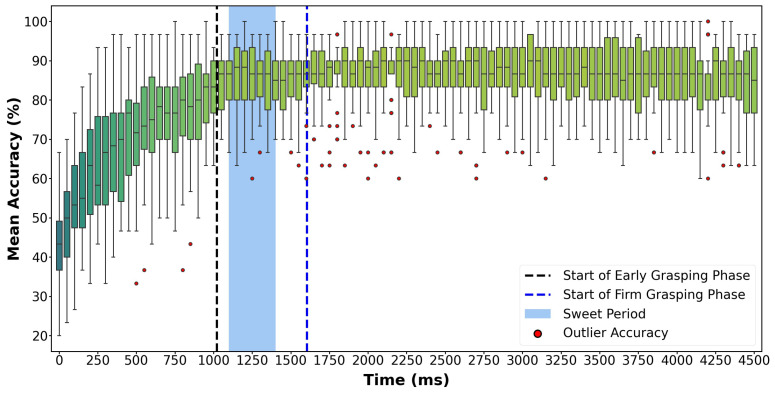
Mean accuracy at each time point during the entire grasp period. This result is from the model which was trained using all three phases data using leave-one-repetition-out cross-validation, and the mean accuracy represents the average accuracy of 30 subjects. The blue region, starts from 1100 ms and ends from 1400 ms, is the sweet period which was confirmed from the first experiment. The vertical dashed lines are averaged starting times of Early Grasping and Firm Grasping phases, which locates at 1020 ms and 1604 ms, respectively. The red dots are outliers.

**Figure 4 biosensors-12-00057-f004:**
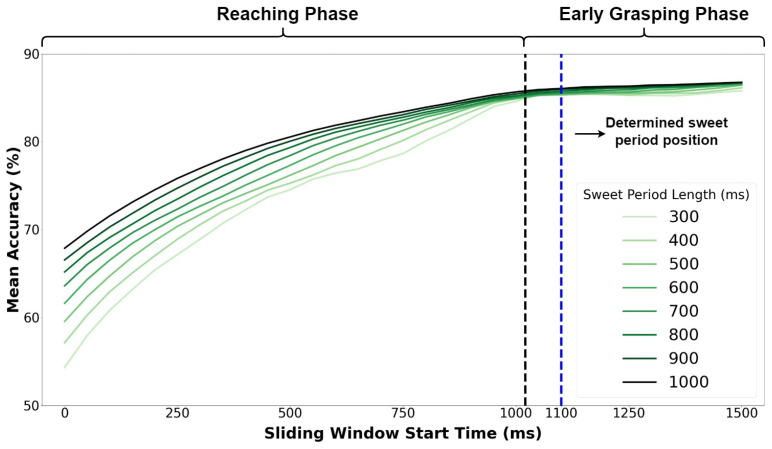
Mean accuracy with different sweet period lengths at different start time.

**Table 1 biosensors-12-00057-t001:** The columns indicate the ID and name of the grasp gestures, the name of the object, and the name of the part of the object involved in the grasping. Adapted from ref. [13].

ID	Grasp Gesture	Object	Grasp Location
		bottle	bottle body
1	medium wrap	can	can body
		door handle	door handle stick
		mug	mug handle
2	lateral	key	key body
		pencil case	case zip
		plate	plate edge
3	parallel extension	book	book body
		drawer	drawer edge
		bottle	bottle cap
4	tripod grasp	mug	mug body
		drawer	drawer knob
		ball	ball body
5	power sphere	bulb	bulb body
		key	key chain
		jar	jar lid
6	precision disk	bulb	bulb body
		ball	ball body
		clothespin	clothespin body
7	prismatic pinch	key	key ring
		can	can pull tab
		remote	remote button
8	index finger extension	knife	knife body
		fork	fork body
		screwdriver	screwdriver body
9	adducted thumb	remote	remote body
		wrench	wrench body
		knife	knife handle
10	prismatic four finger	fork	fork handle
		wrench	wrench handle

**Table 2 biosensors-12-00057-t002:** Window Length Analysis. Both training and test data used the whole grasp period. The classifier used was lightGBM. The features used were STD, RMS, IEMG, MAV, WL, SSI, AAC, and DASDV mentioned in Figure 1. The cross-validation method used was leave-one-repetition-out cross-validation which used one repetition data for testing and the rest three repetitions for training the model, and repeated this process four times to cover all repetitions for testing.

Window Length	Accuracy
50 ms	77.02%
100 ms	78.79%
200 ms	79.98%
500 ms	80.04%
1000 ms	80.33%

**Table 3 biosensors-12-00057-t003:** Analysis Results for Six Cases. All Three Phases include signal from the time of 0 ms to 4500 ms, Firm Grasping Phase is from the time of 2000 ms to 4500 ms, sweet period is from the time of 1100 ms to 1400 ms. Leave-one-repetition-out cross-validation was employed for all cases, such that all testing data was excluded from training the model.

Case Number	Training Data	Testing Data	Accuracy
1	All Three Phases	All Three Phases	79.98%
2	All Three Phases	Firm Grasping Phase	81.68%
3	All Three Phases	Sweet Period	85.50%
4	Firm Grasping Phase	Firm Grasping Phase	80.39%
5	Firm Grasping Phase	Sweet Period	60.80%
6	Reaching Phase	Sweet Period	81.01%
	and Early Grasping Phase		
7	Early and Firm Grasping Phase	Sweet Period	82.51%
8	Sweet Period	Sweet Period	74.99%

## Data Availability

The open source data used in this study is from MeganePro dataset 1 which is available at: https://dataverse.harvard.edu/dataverse/meganepro, accessed on 23 September 2020.

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
