# Peer review of "Phase-Based Grasp Classification for Prosthetic Hand Control Using sEMG"

_biosensors, 2022, doi:10.3390/bios12020057_

Round 1

Reviewer 1 Report

The topic resides in a valuable picture of the "signal extraction-movement control" closed loop. Detecting a sufficiently high resolution in a shorter time may offer a critical time window to drive a prosthetic hand, yet defining and analyzing this time window requires several core aspects such as

(1) how short is sufficiently short?

(2) what is an acceptable signal accuracy?

(3) should this "sweet time" defined differently with different prosthetic types of varying response speed?

Answering these questions necessitates  experiments involving feedback verification based on robotic actuators and comparison/calibration to commercially available sEMG  or FSR, which is absent in the current study leaving its claim "there is a sweet period located in the Early Grasping Phase where sEMG signals can be used to achieve an accurate grasp classification without causing the delay of robotic hand implementation in the real-life applications" poorly verified.   

Author Response

Comment 1. The topic resides in a valuable picture of the "signal extraction-movement control" closed loop. Detecting a sufficiently high resolution in a shorter time may offer a critical time window to drive a prosthetic hand, yet defining and analyzing this time window requires several core aspects such as

(1) how short is sufficiently short?

(2) what is an acceptable signal accuracy?

(3) should this "sweet time" defined differently with different prosthetic types of varying response speed?

Answering these questions necessitates experiments involving feedback verification based on robotic actuators and comparison/calibration to commercially available sEMG or FSR, which is absent in the current study leaving its claim "there is a sweet period located in the Early Grasping Phase where sEMG signals can be used to achieve an accurate grasp classification without causing the delay of robotic hand implementation in the real-life applications" poorly verified.

Response: We appreciate the reviewer’s very thoughtful comments. The authors agree with the reviewer that early detection should be varied among different prothetic arms based on different robotic actuators and calibration system in order to claim "there is a sweet period located in the Early Grasping Phase where sEMG signals can be used to achieve an accurate grasp classification without causing the delay of robotic hand implementation in the real-life applications". However, we do not have a quick answer to question of how short is sufficiently short because each prothetic arm may has different definition. What we did in this  present study was to move this detect time from previous firm grasping phase to the early hand reaching phase. This phase shift in most reaching and grasping tasks will save at least one second in time for the controlling of the prothetic arm.

To support daily activities of disable people using the prothetic arm, we certainly like to achieve the highest level of detection accuracy rate closed to 100%. However in reality, this task is extremely difficult. Current reported accurate rate is about 60-80%. We aimed to match this rate when we move the detection phase from the firm grasp to the early reaching phase.. Thus, we would like to improve the claim to match the actual study findings supported as bellow: " there is a sweet period located in the Early Grasping Phase where sEMG signals can be used to achieve a similar or higher accuracy and lower delay of grasp classification than other time windows, which will improve the performance of robotic hand implementation in the real-life applications." Besides, we also improved the hypothesis in the introduction section “We hypothesize that the muscle activities recorded in the
early period of hand grasping can provide sufficient information to achieve the same
or higher accuracy of grasp classification than other time periods with a
reduced delay for prosthetic hand control.”.

Revisions:

(Disscussion, Page 8, Line 195-198)

Our hypothesis is supported by the results that there is a sweet period located in the Early Grasping Phase where sEMG signals can be used to achieve a similar or higher accuracy and lower delay of grasp classification than other time windows, which would help to improve the performance of robotic hand implementation in the real-life applications.

(Introduction, Page 2, Line 50-53)

We hypothesize that the muscle activities recorded in the
early period of hand grasping can provide sufficient information to achieve the same
or higher accuracy of grasp classification than other time periods with a
reduced delay for prosthetic hand control.

Reviewer 2 Report

The paper deals with the hand gesture recognition problem (grasping), investigating the most suitable phase of the grasping task for identification purposes. The topic is of interest, since reducing the window for feature computation and classification provides significant advantages from the real-time use of such kind of architectures. However, I have some concerns regarding the experimental settings and results which should be carefully addressed. Below you can find my detailed comments.

Please, indicate which classifier was used to obtain the results reported in Table 2, which features, and which cross validation method. The same should be reported also for the results in Fig. 1.

Fig. 2 should be better clarified. It seems that three EMG signals were plotted superimposed each other. Further, the muscular activity is focused during the reaching and early grasping phases. This sounds unusual, since as described in the paper, during the reaching phase almost no forearm muscles activity should be detectable since the arm is approaching the object without touching it. In addition, during the firm grasping phase, the EMG signals appear to have the lowest amplitude, despite during this phase the subject should be actively grasping the object.

lines 128-130: those time values for early and firm grasping phases division (1020 ms and 1604 ms) were set as the same for all the subjects? It sounds quite unlikely that each single subject performed the task with this strictly timing requirements.

lines 146-149: the 2700 (training) and 900 (testing) trials is intended for each of the three different objects grasped and given the 90 pieces of data available for 4.5 s of data? Otherwise, clarify better this aspect.

In Fig. 4 the starting time of the sweet period window on the x-axis is 0 ms, but with respect to the whole signal (reported in Fig. 3) the sweet period window started at which time? 1000 ms or the beginning of the early grasping phase (1020 ms)? Later, since the highest accuracy was reached at 1100 ms, the sweet period was set from 1100 ms to 1400 ms (with a window of 300 ms) and thus it seems that the analysis performed in Fig. 4 started from the beginning of the whole signal. If so, I wonder why it was reported that the sweet period was potentially between 1000 and 2000 ms (lines 157-158) if the subsequent analysis was performed starting from the beginning.

Further, from Fig. 3 it is quite evident that the epoch later marked as the sweet period yet showed the highest accuracy for both the reaching and the early grasping phases. What is the additional value provided by the results summarized in Fig. 4, since the sweet period was yet detectable from the results summarized in Fig. 3?

Please, report the cross-validation method (if any) used for assessing the performances of the classifier used for obtaining the results reported in Table 3. All the available data were used for training and testing the learning model? If so, the accuracy in the cases when training and testing are performed on the same dataset (1,4, and 8) is related to the resubstitution loss?

I wonder whether the highest performances obtained when training the model on all the three phases are related to the fact that a part of the testing data are present in the training dataset. Remarkably, this condition holds in each similar case, except when the model was trained on the firm phase and tested on the sweet phase, where a drop in the accuracy is evident. This could support the above interpretation, since it is the only case where the testing dataset is not a subset of the training dataset.

The fact that using only the so-called sweet period allowed to reach the lowest accuracy should be discussed, since the use of this period can be recognized as one, if not the main, contribution of the paper.

Author Response

 Please note the figures might be missing in this online form, but is available in attached pdf.

Comment 1. Please, indicate which classifier was used to obtain the results reported in Table 2, which features, and which cross validation method. The same should be reported also for the results in Fig. 1.

Response: We appreciate the reviewer’s encouraging comments. For both Table 2 and Fig. 1, the classifier used was lightGBM, the features used were the eight features including Standard Deviation (STD), Root Mean Square (RMS), Integrated EMG (IEMG), Mean Absolute Value (MAV), Waveform Length (WL), Simple Square Integral (SSI), Average Amplitude Change (AAC) and Difference Absolute Standard Deviation Value (DASDV), and the validation method was leave-one-repetition-out cross-validation which used one repetition data for testing and the rest three repetitions for training the model, and repeated this process four times to cover all repetitions for testing. We have added these detailed to the description of Table 2 and Fig. 1.

Revisions:

(Description of Fig. 1, Page 3)

The classifier used was lightGBM. The cross-validation method used was leave-one-repetition-out cross-validation which used one repetition data for testing and the rest three repetitions for training the model, and repeated this process four times to cover all repetitions for testing.

(Description of Table 2, Page 4)

The classifier used was lightGBM. The features used were STD, RMS, IEMG, MAV, WL, SSI, AAC, and DASDV mentioned in Fig. 1. The cross-validation method used was leave-one-repetition-out cross-validation which used one repetition data for testing and the rest three repetitions for training the model, and repeated this process four times to cover all repetitions for testing.

Comment 2. Fig. 2 should be better clarified. It seems that three EMG signals were plotted superimposed each other.

Response: Thank the reviewer for the suggestion. We agree that the previous version of Fig. 2 is not clear enough. Thus, we added the legend to show there are actual 12 channels of sEMG signals from 12 sensors. Some signals amplitude is small and was covered by others.

Revisions:

Comment 3. Further, the muscular activity is focused during the reaching and early grasping phases. This sounds unusual, since as described in the paper, during the reaching phase almost no forearm muscles activity should be detectable since the arm is approaching the object without touching it. In addition, during the firm grasping phase, the EMG signals appear to have the lowest amplitude, despite during this phase the subject should be actively grasping the object.

Response: We appreciate the reviewer’s very careful observations. Although the hand has not grasped to the object during the Reaching Phase, it starts to perform the corresponding grasping gestures related activities such as hand aperture when the sEMG signals from the forearm are usually active with higher amplitude than other phases [26]. In addition, during the Firm Grasping Phase, the muscle status keeps relatively unchanged which makes the amplitude sEMG signal slightly lower than that in the reaching and grasping phase.

Revision: we have further discussed this issue in the discussion section explaining why this phenomenon and how it affects the results (lines 215-225). “Notice that the sEMG signal is more active in the Reaching and Early Grasping phases with high amplitude of the sEMG waveform as shown in Figure 2. This is because the hand starts to perform the corresponding grasping gestures related activities such as hand aperture, where the sEMG signals from the forearm are usually active with higher amplitude than other phases [26], although the hand has not grasped to the object during the Reaching Phase. In contrast, starting from the mid-Early Grasping phase to the whole Firm Grasping Phase, the muscle status keeps relatively unchanged, which makes the amplitude sEMG signal slightly lower than that in the reaching and grasping phase; this is also why better grasp classification performance was achieved during the Early Grasping phase and the Firm Grasping Phase where the sEMG signal patterns are relatively similar.”

Comment 4. lines 128-130: those time values for early and firm grasping phases division (1020 ms and 1604 ms) were set as the same for all the subjects? It sounds quite unlikely that each single subject performed the task with this strictly timing requirements.

Response: The two time values, 1020 ms and 1604 ms for starts of Early Grasping and Firm Grasping phases respectively, were derived from the mean of 30 subjects, each may vary. We will add more detailed to make it more readable (also the caption of figures).

Revisions:

(Materials and method, Page 5, line 124-126)

We segmented the Reaching, Early grasping, and Firm Grasping phases of each grasp gesture from each subject by observing corresponding videos frame by frame and calculated the average value of all the observations.

Comment 5. lines 146-149: the 2700 (training) and 900 (testing) trials is intended for each of the three different objects grasped and given the 90 pieces of data available for 4.5 s of data? Otherwise, clarify better this aspect.

Response: Yes. We improved the corresponding description to: “In other words, there were 324000 data samples (90 samples/grasp * 10 grasp gestures * 4 repetitions * 3 objects * 30 subjects) in the data set”.

Revisions:

(Materials and method, Page 6, line 150-152)

The original text “Therefore, there were nine grasp trials (3 repetitions) in the training data and three (1 repetition) in the testing data for each grasp gesture for each subject. In other words,  there were 2700 grasp trials (900 repetitions) in training data and 900 grasp trials (300 repetitions) in the testing data in total”

was replaced by

In other words, there were 324000 data samples (90 samples/grasp * 10 grasp gestures * 4 repetitions * 3 objects * 30) in the data set.”

Comment 6. In Fig. 4 the starting time of the sweet period window on the x-axis is 0 ms, but with respect to the whole signal (reported in Fig. 3) the sweet period window started at which time? 1000 ms or the beginning of the early grasping phase (1020 ms)? Later, since the highest accuracy was reached at 1100 ms, the sweet period was set from 1100 ms to 1400 ms (with a window of 300 ms) and thus it seems that the analysis performed in Fig. 4 started from the beginning of the whole signal. If so, I wonder why it was reported that the sweet period was potentially between 1000 and 2000 ms (lines 157-158) if the subsequent analysis was performed starting from the beginning.

Response: We appreciate the reviewer’s valuable comments. We agree with the reviewer that the original interpretation of figure 3 and figure 4 might cause confusion. Thus, we have improved to use figure 3 for determining of location of sweet period, and figure 4 for the length of the sweet period. We have revised the manuscript and improve figure 4 accordingly.

Revisions:

(Page 6, line 159-173)

The mean accuracy further increases to relatively stable at around the time of 1250 ms, where we then define the location of the sweet period.

To find the optimal length of the sweet period, we designed different sliding windows with sizes of 300 ms, 400 ms, 500 ms, 600 ms, 700 ms, 800 ms, 900 ms and 1000 ms. The sliding window moved along with the time with step 50 ms, and in each move, it calculated and recorded the mean accuracy. We analyzed the records from the sliding window, and the results are given in Figure 4.

From Figure 4, we can see that the mean accuracy increases with the increase of window length significantly during the Reaching Grasping phase and beginning of the Early Grasping phase (at about 1100 ms) but not significantly afterward. For instance, although the window length of 1000 ms can reach the highest accuracy of 86.3\%, it takes a much longer time than the length of 300 ms with an accuracy of 85.5\%. Therefore, the length of the sweet period is set as 300 ms, and the position is set from 1100 ms to 1400 ms, which makes it entirely located in the Early Grasping phase as the blue region shown in Figure 3.

(Fig. 4, Page 8)

Comment 7. Further, from Fig. 3 it is quite evident that the epoch later marked as the sweet period yet showed the highest accuracy for both the reaching and the early grasping phases. What is the additional value provided by the results summarized in Fig. 4, since the sweet period was yet detectable from the results summarized in Fig. 3?

Response: This comment is address by the response to the previous one.

Comment 8. Please, report the cross-validation method (if any) used for assessing the performances of the classifier used for obtaining the results reported in Table 3. All the available data were used for training and testing the learning model? If so, the accuracy in the cases when training and testing are performed on the same dataset (1,4, and 8) is related to the resubstitution loss?

I wonder whether the highest performances obtained when training the model on all the three phases are related to the fact that a part of the testing data are present in the training dataset. Remarkably, this condition holds in each similar case, except when the model was trained on the firm phase and tested on the sweet phase, where a drop in the accuracy is evident. This could support the above interpretation, since it is the only case where the testing dataset is not a subset of the training dataset.

Response: Thanks again for the reviewer’s comments. We used leave-one-repetition-out evaluation method throughout the study for all evaluation cases including the ones in table 3, which used one repetition data for testing and the rest three repetitions for training the model, and repeated this process four times to cover all repetitions for testing.   This evaluation leave-one-repetition-out method guarantees no testing data are included in the training of the model. For instance, in cases 1, 4 and 8, the training and testing data were from different repetitions and the testing data is not included in the training data, the both training and testing data used the same phase.

Revisions:

Added clarification at (Line 185-189, page 6)

It is worth mentioning again, the cross-validation method used for all the cases was leave-one-repetition-out cross-validation which used one repetition data for testing and the rest three repetitions for training the model, and repeated this process four times to cover all repetitions for testing, such that all testing data was excluded from training the model.

(Caption of Table 3, Page 7)

Leave-one-repetition-out cross-validation was employed for all cases, such that all testing data was excluded from training the model.

Comment 9. The fact that using only the so-called sweet period allowed to reach the lowest accuracy should be discussed, since the use of this period can be recognized as one, if not the main, contribution of the paper.

Response: We appreciate the reviewer’s valuable suggestions. The accuracy is the lowest when using the sweet period data for both training and testing. However, using all phases data for training and the sweet period data for testing achieved the best accuracy, which is main contribution of this paper. The discussion has now been clarified.

Revisions:

(Discussion, Page 9, line 241-246)

For case 8, the accuracy reaches 74.99% only using sweet period for training because this training data also lost the part of information about the gesture in the Reaching Phase and the Firm Grasping Phase. However, using all phases data for training and the sweet period data for testing achieved the best accuracy, which can be the common practice in real-life situations where training a model is not time-sensitive.

Reviewer 3 Report

You tested several methods based in the amplitude-time-space.

Did you check, if there is some information in the frequency-time space or even frequency-amplitude-time space of the signal using a Wavelet Transformation or Wigner-Ville Distribution?

This could improve the paper.

Author Response

Comment 1. You tested several methods based in the amplitude-time-space.

Did you check, if there is some information in the frequency-time space or even frequency-amplitude-time space of the signal using a Wavelet Transformation or Wigner-Ville Distribution?

This could improve the paper.

Response: Thank you for your comments. The authors believe that it is good idea to also use frequency domain or temporal-frequency features. However, the purpose of this study was to identify whether there exists a sweet period before the hand is firmly grasped, instead of exploring more sophisticated features which is out of the scope of this paper.

Round 2

Reviewer 1 Report

I decided to reject this work after going through the revised manuscript. This is based on a criteria I’ve hold for standard research paper published on Biosensors.

Reviewer 2 Report

The authors have addressed all my concerns. I have no further comments regarding the paper.